# A study on the influence of perceived distance on China's inbound Tourism and the interaction of non-economic distance: An analysis of dynamic extended gravity model based on 61 countries' entry data (2004–2018)

**Chengzhe Li[1], Linya He[1], Wei Guo[1], Kewen Wang[1], Sen Tang[2]¤ \***

**1** School of Tourism and Geography Science, Qingdao university, Qingdao, Shandong, China, **2** School of Biology and Geography He Tian Normal College, Hetian, Xinjiang, China

¤ Current address: Department of Marketing, School of Business Administration, Guangzhou Institute of Technology, Guangzhou, Guangdong, China

\* 417238761@qq.com

**Data Availability Statement:** The data of inbound tourism arrivals comes from the https://www.

## Abstract

In the post-epidemic era, the restart of China's inbound tourism is imminent. However, there are gaps in our current understanding of how distance perception dynamically affects inbound tourism in China. In order to understand the past patterns of inbound tourism in China, we mapped the data of 61 countries of origin from 2004 to 2018 into a dynamic expanding gravity model to understand the effects of cultural distance, institutional distance, geographical distance, and economic distance on inbound tourism in China and revealed the dynamic interaction mechanism of non-economic distance perception on inbound tourism in China. Our research results show that cultural distance has a positive impact on China's inbound tourism, while institutional distance has a negative impact. The significant finding is that the dynamic interaction of the above two kinds of perceived distance can still have a positive impact on China's inbound tourism. Its practical significance is that it can counteract the influence of institutional distance by strengthening the cultural distance. Generally speaking, geographical distance and institutional distance restrict China's inbound tourism flow, while cultural distance, economic distance, and interactive perceptual distance promote China's inbound tourism flow.

## Introduction

Affected by the epidemic, China's inbound tourism has operated at a low level in the past three years. The China Tourism Academy predicts that the number of Chinese inbound and outbound tourists is expected to exceed 90 million in 2023, and China's inbound tourism will usher in the main tone of orderly recovery [1]. In January 2023, 《the Tourism Development Plan of the 14th five-year Plan》 issued by the State Council clearly stated that inbound

unwto.org/. You can use https://www.unwto.org/tourism-data/global-and-regional-tourism-performance for part of data of inbound tourism arrivals. Data cannot be shared publicly because of the ethical protections of the UNWTO. If you require complete data, you can contact the unwto Committee at pub@unwto.org.

**Funding:** Funding: This research was funded by National Social Science Found of China, 20Bjy203. The funders had no role in the study design, data analysis, publication decision, or manuscript preparation.

**Competing interests:** The authors have declared that no competing interests exist.

tourism should be promoted step by step. Inbound tourism promotion actions should be launched in due course, and policies should be issued to support the development of inbound tourism, enrich and enhance the national tourism cultural image and promote the high-quality development of inbound tourism [2]. At present, it is difficult to revive the inbound tourism market. How does perceived distance regulate China's inbound tourism market? This question can not only guide the recovery of China's inbound tourism market but also provide some inspiration for the globalization of China's tourism market.

Although Countries' strategies and measures can effectively promote the development of international tourism, international political situation, cultural deviation and COVID-19 also hinder the expansion of the international tourism market. To our knowledge, the main factors of international tourism include the cultural distance, institutional distance, geographical distance and economical distance between destinations and origin countries. The effects of geographical distance and economical distance in tourism demand have been examined, and were negative in most previous studies. However, with the development of the global economic integration and traffic technology, the importance of geographical distance and economical distance has been decreased. And international tourists spiritual demands have been increased, thereby international tourists may prefer destination, which is institutional stable, and has a different culture to their original country. In other words, the cultural distance and institutional distance may play the key role of international tourism. Thus, analyzing the effect of cultural distance and institutional distance in international tourism and exploring the influence mechanism of them have important practical significance for expanding China's inbound tourism market and sustainable development in the current international situation.

As reviewing the previous literature of international tourism, the international tourism demand and factors on inbound tourism is the mainstream of studies. It is obvious that the structure of inbound tourism market and outbound tourism market [3,4] can not be ignored in the demand of international tourism [5,6]. As the accessibility of data, most previous studies centered on the factors of inbound tourism market, such as geographical condition, social economy, culture, and climate. The institutional distance and cultural distance have been the sight of current researchers [7–14], because of the development of transportation and information technology. And the most previous studies analyzed quantitatively by Gravity Model, which is widely used to explore the factors of spatial economic flow. However, the mainstream studies only explored cultural and institutional effect on inbound tourism, and ignored the interaction effect of cultural distance and institutional distance in tourism demand. In addition, the most previous studies taken static analysis, such as POLS, Fixed Effects Model(FE) and Random Effects Model(RE) to explore effect, lacked the dynamic analysis.

The aim of this paper is to dynamically analyze the interaction of cultural distance and institutional distance, and divide their effect in the impact on international tourism demand. Although some previous studies have put cultural distance and institutional distance into international tourism demand model, they have ignored the influence effects between variables and influence mechanisms. Thus, this paper used the panel data from 2004 to 2018 of China and its 61 countries of origin for inbound tourists to examine the effects of culture and institutional distance on inbound tourist arrivals. In addition, based on the expansion gravity model, this paper constructs the dynamic expansion gravity model, and uses the GMM to analyze the dynamic effect of cultural distance (CD), institutional distance (ID) economic distance (ED) and geographical distance (GD) on inbound tourism. It also needs to be emphasized that the dynamic effect on inbound tourism may be important for China's inbound tourism market and foreign strategy.

The mainstream of the previous tourism studies center on effects of the economic factors and geographical factors on international tourism. One of the main contributions of this paper

is to examine the effects of cultural distance and formal institutional distance on inbound tourism. Another contribution of this paper is that, unlike the most previous studies using the static analysis to explore the effects, we use the dynamic expansion gravity model and apply the modified dynamic cultural distance [15] to analysis the dynamic effects on inbound tourism. Lastly, considering the interactive effect of cultural distance and institutional distance on inbound tourism, this paper constructed 4 models to analyze the difference between cultural and institutional effect on inbound tourism, and analyze the interactive effect, which has practical implications for adjustment of the country's international tourism market and draft of foreign strategy. It has important guiding significance for the recovery of China's entry market and the drafting of foreign strategy after the epidemic.

The structure of this paper can be introduced to 7 sections. Section 1 is the introduction to this paper, which introduces the background and the significance of this paper. Section 2 is literature reviews, which reviews the previous studies. Section 3 introduces models and data. Section 4 is the empirical results. Section 5 is the discussion of the results and conclusions. Section 6 outlines the contributions and some implications. Section 7 is the limitations and future prospects.

## Literature review

The factors of international tourism are still popular in previous tourism studies. The mainstream of previous studies analyzed the effects of geographical and economic distance on inbound tourism using the Gravity Model. With the development of international tourism, transportation, and information technology, researchers found that economic and geographical factors may only be some of the factors influencing international tourism, and gave their insights on non-economic factors, such as cultural distance [8,9,12], institutional distance [10,11,13,14], and climate [16–18].

## Theoretical basis

Tourism risk perception theory stresses that the condition of destinations can influence the tourist's choice [19]. With the influence of tourism risk, tourists will choose the destination which is more stable and safer. In other words, tourists will choose to avoid risks and abandon some novel destinations. The most previous studies divided the tourism risk into four dimensions, such as terrorism, war and political instability, health problems and crime [20–22]. The tourism risk can be further divided into cultural risk, equipment risk, financial risk, health risk, physical risk, political risk, psychological risk, satisfaction risk, social risk, time risk, and terror risk [23]. It is obvious that the political risk and cultural risk are important dimensions of tourism risk. As the political and cultural differences between destinations and source areas increase, tourists will choose safer and more stable destinations to visit.

Novelty seeking theory emphasizes that the human prefers to seek the new product or service experience [24,25]. The Novelty seeking can also be named as curiosity drive [26], sensation seeking [27], exploratory drive [28]. One of the motivations of tourist is to experience the novelty of the destination [29]. Novelty seeking theory is regarded as the theoretical basis for tourist destination selection [30,31]. Jang &Feng(2007) found that the level of novelty seeking had a positive effect on tourists' willingness to revisit [31].

## Institutional distance and international tourism

The concept of institutional distance(ID) in tourism is mostly the distance of formal institutions. The institutional distance always describes the difference on political stability, government effectiveness and institutional environment between original country and destination.

The informal distance may conclude cultural distance, physical distance and economic distance. Previous studies about institutional distance and international tourism is really limited. The most previous studies centered on the political stability of destinations on the tourism demand, and found that the institutional quality of destination has a positive effect on inbound tourist arrivals [32]. However, there are limited studies analyzed the effect of political through considering the differences in the political environment between tourism demand and the supply side together. Based on the tourism risk theory, the political risk has a negative effect on tourists' choice of destination. Institutional distance also has a negative effect on China's outbound tourism along"One Belt and One Road" [33]. However, Institutional distance has significant positive effects on Indonesia's inbound tourism in the Association of Southeast Asian Nations(ASEAN) by study of Budisusila [34]. Institutional distance is also a core factor of non-high net worth international tourism flow [35].

## Cultural distance and international tourism

It is well-known that cultural distance is one of the main factors on inbound tourism. However, there is still a dispute about the influence effect on inbound tourism. From the tourism risk perception theory, tourists' perceptions of cultural differences in tourist destinations will create cultural risks and affect their intention to visit. Based on Novelty seeking theory, cultural difference between the original country and destination can meet the novel demands of tourists. Thus, the effect of cultural distance on tourism demand can not be sample seen as a liner.

The mainstream of previous studies about cultural distance can divide into the individual perceived cultural distance and the country's cultural distance. The previous studies about the relationship between perceive cultural distance(PCD) and the tourism intention, mainly used the concepts and dimension of culture which proposed by Hofstede [36,37] and the design questionnaire to analyze the causal relationship between those. Perceived cultural distance has a negative effect on tourism destination's choice [7]. Similarly, it has been found that perceived cultural distance has an insignificant negative effect on Chinese tourists' choice of international tourism destination by Logit analysis and regress analysis [38,39]. However, cultural distance has positive effects on short-haul overnight tourism demand in Hong Kong [8]. In addition, the effect of Japanese perceive cultural distance on the choice of destinations is more significant than Chinese by study of Yang [12].

Some previous studies found that the effect of cultural distance on tourism demand is negative or nonlinear. Bi(2018)& Yin(2020) [38] taken the sample of China inbound tourism, and found that cultural distance has 'cover U shape' effects on inbound tourism. In other words, tourists who are from short haul original countries, prefer to choose a destination with highly cultural distance. And tourists who are from long haul original countries, prefer to choose a destination with lowly cultural distance. However, coming to the opposite conclusion on medical tourism [40]. Cultural distance has negative effects on China's tourism OFDI in Asia [33]. Although cultural distance has negative effects on tourism demand, the elasticity of tourism demand to cultural distance is less [41].

However, the positive effect of cultural distance on international tourism demand has become more significant. As cognitive and effective images of tourist destinations having a positive impact on tourist expectations [42]. Chinese tourists' preferences for tourist destinations led to an increase in the number of tourist arrivals to Western Europe through Disegna's study [43]. Cultural distance leads to great differences in the perception of Taiwan image among mainland tourists, which promotes the actual and potential travel intention [44]. From push and pull motivation, culture is the main pull factor in destination [45]. Thus, some tourists may prefer to visit destinations with highly cultural distance to meet the cultural demand

[46]. And as international tourism is a middle and super tourism market, international tourists have more prominent spiritual needs. Thus, the pursuit of cultural differences will be more intense.

Although the role of culture and institution on international tourism is really important, there are only handfuls of empirical studies [5,7,8,34,39,40,42,47]. The most previous studies centered on the effect of cultural distance on inbound tourism, and found that cultural distance has a negative effect on inbound tourism or tourism demand. But the data of independent variable cultural distance previous studies using is Static, which can not objectively reflect the cultural distance between countries in different periods. And the spiritual demands of international tourists have not been focus seriously either. Especially, as the improvement of the Internet and transport, tourists can get enough information about the destination before travel, thus tourists' perceive tourism risk of destination has been significantly decreased. And the attraction of cultural difference has been significantly increased. The previous studies have not taken focus on the interaction of cultural distance and institutional distance. However, the cultural distance and institutional distance can affect the international tourism directly, the effect is not separate.

The effect of interaction of cultural distance and institutional distance on international tourism can not be ignored. Based on international tourism influencing factors, the cultural distance and institutional distance are non -economical distance. The effect of cultural distance is mainly reflected in the impact of tourist values and customs, while the effect of institutional distance tends to the specifications of tourists' behavior and system. However, culture and institution are not completely independent variables. In essence, regional cultural differences are an important foundation for regional institutional differences, and institutions are an exterior embodiment of culture.Therefore, exploring the interaction effect of cultural distance and institutional distance can better explain the difference between the impact of cultural distance and institutional distance on international tourism.

In addition, the cultural environment and institutional environment are important factors of international tourism flow. In particular, under the limited international economy, the soft power of culture, politics, diplomacy and other countries is an important measure for countries to promote international tourism development and expand and stabilize the international tourism market. However, the use of cultural strategy, diplomatic strategy, or a comprehensive cultural diplomatic strategy is still an important problem that plagues the international tourism market strategy choice. The model of international tourism demand would be more completed by the adoption of interaction of cultural distance and institutional distance.

## Materials and methods

This section may be divided by subheadings. It should provide a concise and precise description of the experimental results, their interpretation, as well as the experimental conclusions that can be drawn.

### Model and data

To explore the effect of cultural distance and institutional distance on inbound tourism, we applied the dynamic expansion gravity model, which is based on the commodity version of the Gravity Model. It can examine the spatial effect between countries, and has been generally applied to analyze geographical flow(migration and tourism). The Gravity Model is specified as

$$Q = \frac{m_i m_j}{d^2} \tag{1}$$

where Q stands for the number of tourism arrivals at destination, m is the characteristic of the country, and d reflects the geographical distance between the original country and the country of destination, i and j is the country of origin and the country of destination, respectively.

In order to examine interactive effects by dynamic analysis, this paper introduces variables $Tour_{cjt-1}$ and $CD^*ID_{cjt}$ into the empirical model, and shown as following:

$$Tour_{cit} = \beta_0 + \beta_1 Tour_{cit-1} + \beta_2 CD_{cit} + \beta_3 ID_{cit} + \beta_4 CD*ID_{cit} + counting + \mu_i + \varepsilon_{cit} \qquad (2)$$

where dependent variable Tour stands for the number of tourist arrivals of China inbound tourism from original countries. The independent variables include $Tour_{cjt-1}$, $CD_{cjt}$, $ID_{cjt}$ and $CD^*ID_{cjt}$. The c, j, t stands for China, the original country j and time period respectively. Last, the $\mu_c, \varepsilon_{jt}$ and counting represents the error term and control variables respectively.

The CD stands for cultural distance in this model. The most previous studies applied Kogut &Singh's cultural index [48]and six cultural dimension scores [37] to measure cultural distance [8]. Based on the demand of dynamic analysis, diplomatic years between China and the sample countries are introduced into the cultural distance formula as the data of bilateral cultural exchanges to carry out dynamic processing. This paper applied the revised model proposed by Qi [15] as:

$$CD_{cjt} = \{\sum_{n=1}^{6} [(I_{cn} - I_{jn})/V_n]/6\} + (1/Y_{cjt}) \qquad (3)$$

where $CD_{cjt}$ is the cultural distance of China from the j country in the time of t, $I_{cn}$ the Hofstede's score for the nth cultural dimension of China, $I_{jn}$ the Hofstede's score for the nth cultural dimension and jth country, $V_n$ the variance of the index on the nth dimension, and n the number of cultural dimensions. $Y_{cjt}$ is the years of the diplomatic relation between China and jth country in time of t.

The ID stands for institutional distance.These data are collected from WGI, including civic rights, political stability, government efficiency, management quality, laws and regulations, and anti-corruption capabilities. And the Kogut &Singh's (1988) [48] Variance measurement is applied in calculating the institutional distance as

$$ID_{cjt} = \{\sum_{n=1}^{6} [(I_{cn} - I_{jn})/V_n]/6\} \qquad (4)$$

where $ID_{cjt}$ stands for the institutional distance of China and country j in the period t. $I_{cn}$ and $I_{jn}$ is the WGI score for the nth institutional dimension of China and country j respectively. $V_n$ is the variance of the index on the nth dimension. And n is the number of institutional dimensions.

$CD^*ID_{cjt}$ is the interaction of $CD_{ijt}$ and $ID_{ijt}$, which can be used to examine the interactive effect. $Tour_{cjt-1}$ stands for the first order lag of Tour, the independent variable, which can examine the dynamic effect of tourism.

The controlled variables in this paper include geographical distance, economical distance, boarder, language and T2009. The geographical distance is widely accepted with negative effects on inbound tourism. GD size in this paper is based on the distance among the capitals of the countries in CPEII database. The economical distance in this paper is the per capita GDP(PGDP) ratio between China and origin countries, which has a positive effect on inbound tourism. The PGDP in this paper is obtained from the World Development Indicators of the World Bank. T2009, Language and boarder in this paper are virtual variables. The T2009 responds the effect of financial crisis in 2009, which has a negative effect on international tourism. Language is the language similarity between China and the original countries, which is widely accepted with a positive effect on tourism. Boarder in this study is whether the original

country border China, which reflects the cost and destination accessibility and has a negative effect on inbound tourism.

The number of entry -travelers and geographical distance variables in this article are relatively large, while the values of variables such as cultural distance and political distance are small, and standard units are used. Due to the differences in the unit and value in the study, the degree of fitting and credibility of the research results will be reduced. In order to reduce the impact of the data differences on the results of the research model, this paper applies a logarithm, which can help to eliminate trends [49], and to avoid the difference of the unit of data, on variables Tour and GD. And Eq 2 is augmented as

$$LnTour_{cit} = \beta_0 + \beta_1 LnTour_{cit-1} + \beta_2 CD_{cit} + \beta_3 ID_{cit} + \beta_4 CD*ID_{cit} + \beta_5 LnGD_{cit} + \beta_6 ED_{cit}$$
$$+ \beta_7 T2009 + \beta_8 boarder + \beta_9 language + \mu_i + \varepsilon_{cit} \tag{5}$$

Affected by the COVID-2019, Chinese inbound tourism data for 2019–2022 is missing and belong to abnormal data.For this study, abnormal data processing has relatively little effect on the conclusions and empirical significance of this study, and will further the sample selection of this paper, resulting in a decrease in sample representation of this paper. Therefore, this paper selects variables such as Chinese inbound tourist arrivals, cultural distance and institutional distance from 2004 to 2018 for research. The specific variable definitions and data sources are shown in Table 1.

Based on the availability of data, this paper collects a total of 73 countries' cultural distance data, the institutional distance data of 87 countries. And considering the integrity and representation of data, the study uses 61 countries of origin in China inbound tourism statistic. The 61 countries of origin include the different regions of the world. And those countries have good represent of China's inbound tourism market, whose inbound tourist arrivals are more than 70 percents in overall China's inbound tourist arrivals. And the data are shown in Table 2.

The descriptive statistics of variables in this paper are shown in Table 3. The lowest and highest value of the dependent variable Ln-Tour(LnTour) is -1.958 and 6.632, respectively. LnTour's mean and standard deviation is 2.031 and 1.832, respectively. It can also find that the mean of independent variables CD, ID and CD*ID is 2.584, 1.856 and 0.760. The mean of control variables Ln-geographical distance(LnGD) and ED is 8.888 and 4.626. The control variable BOARDER, LANG and T2009 are dummy variable, have the highest and lowest value of 1 and 0 respectively.

**Table 1. Definition of variables and data sources.**

| Variables | Definition | Data Source |
|---|---|---|
| Dependent variable | | |
| Tour | The China's inbound tourist arrivals from the Country j | UNWTO |
| Independent variable | | |
| CD | Cultural distance between China and the Country j | Hofstede |
| ID | Institutional distance between China and the Country j | WGI |
| Control variable | | |
| ED | Economic distance between China and Country j | World bank |
| GD | Geographical distance between China and Country j | CEPII |
| T2009 | Financial crisis in 2009 | |
| Boarder | Whether Country j border China. | CEPII |
| Language | Whether Country j has language similarity to China. | CEPII |

**Table 2. The overview of countries of original.**

| Regions | Country | Obs |
|---|---|---|
| North America | United states, Canada, Mexico, Colombia | 4 |
| Europe | Germany, Austria, Czech Republic, Switzerland, Norway, Denmark, Belgium, United Kingdom, France, The Netherlands, Greece, Portugal, Spain, Italy, Belarus, Ukraine, Lithuania, Latvia, Slovakia, Poland, Hungary, Romania | 22 |
| Asia | Japan, North Korea, Singapore, Vietnam, Bengal, Indonesia, Malaysia, Philippines, Thailand, India, Russia, Pakistan, Kazakhstan, Azerbaijan, Iran, Egypt, Jordan, Saudi Arabia, Iraq, Lebanon | 20 |
| South America | Peru, Brazil, Bolivia, Chile, Argentina, Venezuela | 6 |
| Africa | Zambia, Ghana, Morocco, Nigeria, South Africa, Algeria, Libya | 7 |
| Australia | Australia, New Zealand | 2 |

## Methodology

As non-stationary time-series variables lead to spurious regression, which can affect the accuracy of results, the first step of this paper is to examine the stability of the LnTour, CD, ED, ID by the panel Levinlin test(LLC) and Im-Pesaran-Shin test(IPS).

Then, as the variables selected may be similar or have a strong correlation that can cause endogenous problems and influence the estimated accuracy, we need to analyze the correlation between variables by Pearson test, before official regress.

As the geographical distance in this paper is a time-invariant variable, which cannot be analyzed by FE. And the pooled OLS regression may be problematic, because of the heteroskedasticity and auto-correlation, which rejects the null hypothesis of the Breusch–Pagan test (BP) and Wald test(Chi2 = 0.68, p = 0.4103; Chi2 = 25.92, p<0.001). Although the OLS estimators are unbiased and consistent, they are not efficient, and the standard errors are not rectified [50]. Considering that the CD and LnTour may influence each other, which results in an endogenous problem, this paper adopts a dynamic model for analysis.The dynamic panel model can use differential GMM estimation(DGMM) and system GMM estimation(SGMM). The following dynamic models shown as

$$y_{it} = \alpha + \rho y_{i,t1} + x'_{it}\beta + z'_i\delta + \mu_i + \varepsilon_{it} \quad (t = 2, \ldots, T) \tag{6}$$

First, make a first-order difference to eliminate individual effect μi, we can get

$$\Delta y_{it} = \rho y_{i,t\,1} + \Delta x'_{it}\beta + \Delta \varepsilon_{it} \quad (t = 2, \ldots, T) \tag{7}$$

**Table 3. Descriptive statistics of variables.**

| Variable | Obs | Mean | Std. Dev. | Min | Max |
|---|---|---|---|---|---|
| LnTour | 915 | 2.031 | 1.832 | -1.958 | 6.632 |
| CD | 915 | 2.584 | 0.979 | 0.562 | 4.956 |
| ID | 915 | 1.856 | 1.640 | 0.084 | 6.318 |
| CD*ID | 915 | 0.760 | 1.916 | -4.151 | 10.584 |
| LnGD | 915 | 8.888 | 0.551 | 6.862 | 9.868 |
| ED | 915 | 4.626 | 6.083 | 0.139 | 38.182 |
| BOARDER | 915 | 0.082 | 0.274 | 0 | 1 |
| LANG | 915 | 0.033 | 0.178 | 0 | 1 |
| T2009 | 915 | 0.067 | 0.250 | 0 | 1 |

However, $\Delta y_{i,t-1}$ is still related to $\Delta\varepsilon_{it}$. Therefore, $\Delta y_{i,\,t-1}$ are endogenous variables, and it is necessary to find suitable instrumental variables for consistent estimation. Therefore,it can use $y_{i,\,t-2}$ as the instrumental variables of $\Delta y_{i,\,t-1}$, and then perform 2SLS estimation(differential GMM estimation, DGMM). However, DGMM estimation has certain limitations. First, the premise of DGMM is that the disturbance term does not have auto-correlation(Cov $(\varepsilon_{it},\varepsilon_{is}) = 0$, $t{\neq}s,{\cup}i$). Secondly, DGMM cannot estimate the variable $z_i'$ that does not change with time; when the explained variable has strong continuity(when it tends to 1, $y_{i,\,t-2}$, etc, are no longer correlated with $\Delta y_{i,\,t-1}$). So it is no longer a valid instrumental variable, as the problem of weak instrumental variables. To solve the above problem, while estimating the difference Eq (7), use the first-order difference lag term $\{\Delta y_{i,\,t-1}, \Delta y_{i,\,t-2}, \Delta y_{i,\,t-3},\ldots\}$ shown as the Eq (6). The instrumental variables of Eq (6) are estimated, which is the "systematic GMM method".

## Results

### Panel unit root test

We use the LLC test and IPS test to analyse the stability of the panel data. The result, related to the test, is reported in Table 4. The LnTour, CD and ED are stationary by LLC Test and IPS test in an original sequence. The variable ID and CD*ID are stationary by LLC test, but are not stationary by IPS test in the original sequence. All variables are stationary by LLC test and IPS test through one order difference. Therefore, the variables of this paper are relatively stationary. As the data of this paper is from 2004–2018, relatively short, we still use the original sequence to analyze.

### System GMM analysis

The paper analyzes the effect of cultural distance and institutional distance on inbound tourism by system GMM analysis. When some variables in the model are endogenous, systematic GMM analysis can eliminate endogenous bias and provide more efficient estimation results. Therefore, this paper introduces the first-order lag term of LnTour(LnTour L1.) as an explanatory variable. In addition, hansen test can effectively test the validity of instrumental variables. The estimated result is shown in Table 5. Mode 1 to Model 4 analyzes the effect of CD, ID, CD and ID, and the overall effect respectively. As Table 6 shows, the number of objectives in all models is 854. And AR(1) and AR(2)in all model is significant at the 5% level and insignificant, respectively. It means that the model of this paper accepts the assumption that the first-order auto-correlation coefficient of the disturbance term difference is not zero and the second-order auto-correlation coefficient is zero. Therefore, it is acceptable to assume that the disturbance term does not have auto-correlation, in other words, the model can be estimated by

**Table 4. Panel unit root test.**

| Variables | Original sequence | | One order difference | |
|---|---|---|---|---|
| | LLC test | IPS test | LLC test | IPS test |
| LnTour | -6.969*** | -8.845*** | -12.651*** | -12.041*** |
| CD | -32.501*** | -23.389*** | -50.299*** | -22.241*** |
| ID | -5.312*** | 0.290 | -6.594*** | -12.391*** |
| ED | -14.128 *** | -3.536*** | -6.736*** | -6.880*** |
| CD*ID | -6.746*** | 0.196 | -6.504*** | -12.220*** |

Note

*** $p < 0.01$.Null hypothesis:The sequence xt is non-stationary.

**Table 5. The result of system GMM.**

| variable | Model 1 | Model 2 | Model 3 | Model 4 |
|---|---|---|---|---|
| LnTour L1. | .935*** | .936*** | .878 *** | .860*** |
|  | (.023) | (.018) | (-.049) | (.061) |
| CD | .008 | \ | .191* | .102 |
|  | .059 |  | (.108) | .131 |
| ID | \ | -.081** | -.125** | -.408*** |
|  |  | (.035) | (.051) | (.151) |
| CD*ID | \ | \ | \ | .093* |
|  |  |  |  | (.055) |
| LnGD | -.172* | -.235*** | -.483** | -.533** |
|  | (.077) | (.089) | (.216) | (.223) |
| ED | .010*** | .019*** | .016*** | .014*** |
|  | (.002) | (.005) | (.003) | (.005) |
| T2009 | -.111*** | -.110*** | -.113*** | -.117*** |
|  | (.022) | (.021) | -0.02 | (.020) |
| LANG | -.105 | .350 | 1.175 | 1.346 |
|  | (.568) | (.695) | (1.419) | (1.563) |
| BORDER | .071 | -.055 | .055 | .014 |
|  | (.087) | (.104) | (.162) | (.199) |
| Constant | 1.669*** | 2.354*** | 4.233** | 4.962** |
|  | (.614) | (.830) | (1.822) | 1.896 |
| Obs | 854 | 854 | 854 | 854 |
| AR(1) | -2.37 | -2.4 | -2.4 | -2.44 |
| P-value | 0.018 | 0.017 | 0.017 | 0.015 |
| AR(2) | -0.92 | -0.88 | -0.84 | -0.79 |
| P-value | 0.356 | 0.381 | 0.403 | 0.432 |
| Hansen test | 59.97 | 60.08 | 59.8 | 59.87 |
| P-value | 0.209 | 0.265 | 0.243 | 0.212 |

Note

*** $p < 0.01$

** $p < 0.05$

* $p < 0.1$.

GMM. In addition, the results of Hansen' test in model 1 to model 4 show that all instrumental variables are valid.

From the view of the variable's effect on inbound tourism in different models, CD has positive effects on inbound tourism in model 1, 3, and 4, but is only insignificant in Model 1. In addition, the influence effect in model 3 and 4 is significantly higher than model 1. The ID has a significant negative effect on inbound tourism in model 2, 3 and 4. The effect in model 3 and 4 is higher than model 1. The CD*ID has significant positive effects on inbound tourism in model 4. The control variable LnGD and T2009 can be regarded as hindering factors on inbound tourism in all models. LnTour L.1 and ED have significant positive effects on inbound tourism in all models. The effect of Language and Boarder is not significant.

From the model 1, we can find that the effect of CD is insignificant and weak (0.008, $p > 0.1$). LnGD and ED (-0.173, $p < 0.1$; 0.10, $p < 0.01$) are the main factors of China's inbound tourism. At the same time, from the model 2, we can find that the main factors of inbound tourism is geographical distance(-0.235, $p < 0.01$), but the effect of institutional

**Table 6. Robustness test.**

| Variable | Model 1 | Model 2 | Model 4 | | | Model 1 | Model 2 | Model 4 |
|---|---|---|---|---|---|---|---|---|
| | | | **Including Vietnam** | | | **Excluding Vietnam** | | |
| | **CD** | **ID** | **CD** | **ID** | **ID&CD** | **CD** | **ID** | **ID&CD** |
| | **EDI** | **EDI** | **EDI** | **EDI** | **EDI** | **Hofstede** | **Hofstede** | **Hofstede** |
| LnTour L1. | .935*** | .941*** | .905*** | .865*** | .911*** | .911*** | .934*** | .844*** |
| | (.020) | (.017) | (.037) | (.067) | (.036) | (.021) | (.016) | (.067) |
| CD | .001 | / | .001 | .089 | .001 | .078 | / | .151 |
| | (.003) | | (.005) | (.204) | (.007) | (.058) | | (.159) |
| ID | / | -.074** | -.578 | -.316** | -.386* | / | -.065* | -.352* |
| | | (.037) | (.248) | (.143) | (.215) | | (.034) | (.181) |
| CD*ID | / | / | .005 | .061 | .003 | / | / | .076 |
| | | | (.003) | (.055) | (.002) | | | (.063) |
| Control | Yes | Yes | Yes | Yes | Yes | Yes | Yes | Yes |
| Constant | 1.683*** | 2.408*** | 3.835*** | 5.447** | 4.091*** | 2.019** | 2.014* | 5.015** |
| | (.570) | (.913) | (1.379) | (2.316) | (1.555) | (.705) | (.737) | (2.137) |
| Obs | 854 | 854 | 854 | 854 | 854 | 840 | 840 | 840 |
| AR(1) | -2.37 | -2.4 | -2.43 | -2.45 | -2.44 | -2.34 | -2.35 | -2.45 |
| | .018 | .016 | .015 | .014 | .015 | .019 | .019 | .014 |
| AR(2) | -0.92 | -1.08 | -0.82 | -1.19 | -1.13 | -0.82 | -0.79 | -0.8 |
| | .357 | .282 | .412 | .234 | .258 | .414 | .432 | .424 |
| Hansen test | 60.09 | 60.16 | 60.04 | 59.87 | 60.02 | 59.03 | 59.3 | 59.14 |
| | .265 | .263 | .207 | .212 | .208 | .297 | .288 | .261 |

Note

*** $p < 0.01$

** $p < 0.05$

* $p < 0.1$.

distance on inbound tourism(-0.081, $p < 0.01$) is more than effect of economic distance(0.019, $p < 0.01$). When we take the CD and ID in one model(Model 3), the effect of CD and ID improves obviously, especially for CD.

As the result of model 3 shows, the coefficient associated with CD is($-0.191$, $p < 0.1$) indicating that China's inbound tourist arrivals prefer culturally diverse destinations. The coefficient associated with ID is indicating that China's inbound tourist arrivals prefer destinations with similar political systems and institutional environments($-0.125$, $p < 0.05$). The effect of CD and ID is obviously higher than the effect of economic distance($-0.016$, $p < 0.01$). In other words, with the development of transportation and information technology, no-economic factors, such as culture, and institutional difference, may be the main factors of China's inbound tourism. Compared to the coefficient associated with cultural distance and institutional distance in model 1 to 3, it is clear that there may be an interactive effect between cultural distance and institutional distance. Thus, we conduct the model 4 to examine the interactive effect.

As the model 4 shows, CD*ID has significant positive on China's inbound tourism(0.093, $p < 0.1$). And the effects of CD and ID on inbound tourism have little improvement(0.102, $p > 0.1$; -0.168, $p < 0.01$). Compared with the model 2, 3 and 4, cultural distance plays a moderating role in the impact of institutional distance on inbound tourism demand. And the moderating effect of cultural distance on the relationship between institutional distance and tourism demand is positive.

## Robustness test

Robustness test can be conducted by three methods: comprehensive or separate transformation estimation method, sample elimination and change variable index. This paper adopts the methods of eliminating part of the samples and changing the variable index. Vietnam is an important neighboring country and one of the source countries of inbound tourism, but after 2014, China's statistical caliber of inbound tourism talents in Vietnam has changed, which will have a certain impact on the overall estimate of this study. In order to exclude the influence of statistical caliber differences on the research results, the Vietnam sample is excluded from the total sample. Euclidean distance (EDI) and Hofstede are popularly used in the measurement of cultural distance and institutional distance. So index of cultural distance and institutional distance used the EDI.

As Table 6 shows, cultural distance still has positively promoting effect on inbound tourism. Institutional distance still has a significant negative impact on China's inbound tourism. And, the interaction of cultural distance and institutional distance still has a significant positive effect on inbound tourism. All these indicate that the results of this study are robust.

## Discussions and conclusion

### Discussions

This paper uses the system GMM analysis and conducts 4 models to examine the dynamic effect of institutional distance and cultural distance on China's inbound tourism. And it can be found that the institutional distance has a negative effect on China's inbound tourism. Compared with the results of 4 models, cultural distance can be seen as a moderating variable in China's international tourism demand model.

It is clear that the introduction of cultural distance and the interaction of cultural distance and institutional distance in the tourism demand model has positively improved the effect of institutional distance on inbound tourism. It means that the moderating effect of cultural distance on the relationship between institutional distance and inbound tourism is negative.

Unlike the previous studies, this paper introduces a dynamic cultural distance to examine the effect on inbound tourism. The results of cultural distance on China's inbound tourism may be counter to some studies [40,41,47], but is similar to others [43,44]. In addition, the data of this paper may be another reason for this result. The most countries of origin selected in this paper is in America, Europe and Africa, where the cultural distance and geographical distance is relatively higher. And cultural distance has positive effects on long-haul travel [47]. The cultural difference between the country of origin and the country of destination can be attractive for inbound tourist arrivals. Lastly, the inbound tourist arrivals can be affected by the number of inbound tourist arrivals in in origin countries. The reason for the promotion effect is that a good travel experience for tourists will introduce China to their friends and relatives. China can attract more international tourists by building a good national image.

The hindering factors of China's inbound tourism are geographical distance, institutional distance and T2009. Although the main hindering factor of China's inbound tourism is geographical distance, the effect of institutional distance cannot be ignored. It is well known that the geographical condition is the main factor of tourism, which affects the cost of transportation and accessibility of destination, especially for international tourism. In addition, the policy stability of destination and institutional difference has an important influence on choices of international tourists. As institutional differences increase, the travel risk of international tourists will increase rapidly. Thus, international tourists will visit the destination of lowly institutional difference.

## Conclusion

In order to examine the interactive effect of institutional distance and cultural distance on China's inbound tourism dynamically, this paper collected data from 61 countries of China's original countries from 2004 to 2018, and used the system GMM analysis. The results of this paper show that institutional distance has significant negative effect on inbound tourism. Cultural distance has a negative moderating effect on the relationship between institutional distance and inbound tourism. The result also shows that the economic distance and geographical distance are promoting factors and hindering factors of China's inbound tourism, respectively. The result of this paper cannot only give some valuable insight into China's international tourism market but also have significance for pulling the regional cultural exchange and economic cooperation.

## Contribution

Affected by the epidemic, the full recovery of the tourism industry needs a process, and it is not easy to restart or even revitalize the inbound tourism market which has been seriously affected by the epidemic. Since inbound tourism involves the management of more market mechanisms such as visa processing, flight supply, and price adjustment, the preparation cycle and response time of the inbound market are longer. The recovery of the inbound tourism market still faces difficulties such as the reconstruction of the supply chain system, the reconstruction of the business team, and the re-linking of channel users. At present, we should actively attract international tourists, further relax the restrictions on inbound tourism, and implement the inbound tourism revitalization plan. This paper puts forward the following suggestions for the development of China's inbound tourism from the perspective of the international situation and China's international tourism marketing.

Based on the international conditions and China's international tourism marketing, the effect of institutions can not be ignored. The world tourism industry has entered the stage of recovery, and various countries have introduced measures to revitalize the inbound tourism market. Therefore, reducing the institutional distance between China and its origin countries can not only expands China's inbound tourism market but also promotes the development of countries' relationship. Although the political system of the countries can not be changed, countries can still reach a consensus on foreign trade system, international tourism law and international law. Thus, China needs to actively participate in the improvement and practice of the international tourism systems, regulations and international law. China needs to promote political ideas such as A community with a shared future for mankind, to enhance the right to speak internationally, and to establish a responsible image of a big country to attract more international tourists. In addition, China should open up further, and strengthen regional economic and political cooperation such as One Belt One Road, Double Circle and Regional Comprehensive Economic Partnership Agreement (RCEP).

From the view of China's international tourism marketing, cultural distance is the main factor of inbound tourism. It can not only promote the development of the international tourism market but also has a negatively moderating effect on the relationship of institutional distance and inbound tourism. Thereby, the improvement of China's culture can play a key role of the inbound tourism market. It is clear that the international tourism industry in China is still in the exploratory stage, and the integration of culture and tourism cannot meet the international tourism demand. Previously, cities such as Beijing, Shanghai, Chengdu and Xi'an have become well known to international tourists, thanks to the fact that the destination has created a series of continuous foreign tourism marketing activities around inbound tourism innovation, enriching the city tourism brand and image through cultural IP. Therefore, China needs to dig

further the core of traditional culture and build a cultural tourism brand with Chinese characteristics. Moreover, China ought to respect the cultures of all regions and strengthen regional cultural exchanges in order to promote political mutual trust among countries and promote the development of international tourism.

The result of this paper also implies the importance of cultural exchange and political cooperation for regional cooperation and globalization. Under the condition of anti-globalization and trade protectionism, enhancing regional cooperation and peace between countries by regional economic and trade cooperation does not seem to be very effective. Thereby, by using national characteristics and culture to attract international tourists and cultural exchanges, the flow of talents and elements in the region can be further strengthened. As the model in this paper shows, cultural distance and interaction have positive effects on inbound tourism. It means that considering the cultural attraction, tourists may choose to visit some destinations that are politically different from their original countries. Therefore, countries should actively explore the connotations of their own cultures, promote local cultures, attract international tourists, strengthen regional cultural exchanges, and promote regional economic and political cooperation.

## Limitations and future prospects

There are some limitations to giving new insights to future researchers. This paper only focused on the effect of cultural distance, institutional distance and interaction on inbound tourist arrivals, but did not analyze the non-linear effect of cultural distance. The future study can conduct a model to analyze the effect of interaction between the square of cultural distance and institutional distance. Moreover, this study only analyzed dynamically the effect on inbound tourist arrivals by system GMM. The future study can analyze the dynamic effect on outbound tourism. In addition, due to data limitations, only 61 major inbound tourist source countries of China are selected as samples for this study, which is relatively small. The future study can expand the sample size of the study in the future to verify the universality of the model. This paper mainly focuses on language, territorial border and financial crisis in the selection of control variables, and does not include population, exchange rate and transportation in the analysis of control variables. In the future, studies could further consider the effects of variables such as population, exchange rates and transportation.

## Supporting information

**S1 Data.**
(XLSX)

## Author Contributions

**Conceptualization:** Chengzhe Li, Sen Tang.

**Data curation:** Chengzhe Li, Wei Guo, Sen Tang.

**Formal analysis:** Chengzhe Li, Linya He, Kewen Wang, Sen Tang.

**Funding acquisition:** Chengzhe Li, Wei Guo.

**Investigation:** Linya He.

**Methodology:** Kewen Wang.

**Project administration:** Chengzhe Li, Wei Guo, Kewen Wang.

**Resources:** Chengzhe Li, Wei Guo.

**Software:** Wei Guo, Kewen Wang.

**Supervision:** Chengzhe Li, Linya He, Wei Guo, Kewen Wang.

**Validation:** Chengzhe Li, Linya He.

**Visualization:** Chengzhe Li, Linya He, Sen Tang.

**Writing – original draft:** Sen Tang.

**Writing – review & editing:** Chengzhe Li, Linya He, Kewen Wang, Sen Tang.

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
