## [Decision Letter · Decision Letter 0]

30 Aug 2023

PONE-D-23-18100A study on the influence of perceived distance on China's inbound Tourism and the interaction of non-Economic distance: an Analysis of dynamic extended Gravity Model based on 61 countries' entry data (2004-2018)PLOS ONE

Dear Dr. Tang,

Thank you for submitting your manuscript to PLOS ONE. After careful consideration, we feel that it has merit but does not fully meet PLOS ONE’s publication criteria as it currently stands. Therefore, we invite you to submit a revised version of the manuscript that addresses the points raised during the review process.

*Please provide a final paper with all revisions made and I recommend an additional check on plagiarism and/or compliance with the Journal's guidelines.*

We look forward to receiving your revised manuscript.

Kind regards,

Vincenzo Basile, PhD

Academic Editor

PLOS ONE

Reviewers' comments:

Reviewer's Responses to Questions

**Comments to the Author**

1. Is the manuscript technically sound, and do the data support the conclusions?

Reviewer #1: Partly

Reviewer #2: Partly

2. Has the statistical analysis been performed appropriately and rigorously? 

Reviewer #1: Yes

Reviewer #2: Yes

3. Have the authors made all data underlying the findings in their manuscript fully available?

Reviewer #1: Yes

Reviewer #2: Yes

4. Is the manuscript presented in an intelligible fashion and written in standard English?

Reviewer #1: Yes

Reviewer #2: Yes

5. Review Comments to the Author

Reviewer #1: This paper focuses on understanding the impact of different types of distance perception on inbound tourism in China. The paper utilizes a dynamic expanding gravity model to analyze the effects of cultural, institutional, geographical, and economic distance on inbound tourism patterns from 2004 to 2018. The research is well-designed and well-analyzed. However, there are still some major flaws.

1. More control variables are needed. Relevant tourism-related variables such as population and transportation should be incorporated to account for comprehensive influencing factors.

2. The endogeneity of variables needs to be considered. For instance, a decrease in economic distance could lead to an increase in tourist numbers, but an increase could also lead to a decrease in economic distance.

3. Some English writing in the article requires attention, such as adding plural to 'literature reviews'.

4. Specific references in the article also need careful attention to format errors and omissions, requiring correction.

Reviewer #2: Thanks for writing this interesting paper that analyses the implications of different types of "distance" on tourism industry of China. Although this paper has strengths, it also has weaknesses. Therefore, I would like the authors to address these weaknesses so that this paper can capture a wider readership.

The weaknesses are:

1. Sampling. Why only these 61 countries are included in this study? This study uses panel data. We should note that the choice of countries may affect the results. I hope the authors can address this comment carefully.

2. Duration. Any particular reason 2019 data points are omitted since it is a year prior to Covid-19?

3. Typo and grammatical mistakes. I hope the authors can check this paper again to remove obvious typo and grammatical mistakes. For example, line 39 of page 3, there are duplicated "and".

4. Interaction effect between cultural distance and institutional distance. The authors state that the mentioned interaction effect is not considered in existing studies. Why is the theoretical foundation of including this interaction effect? We should not estimate model just based on data availability.

5. Consistency of notations. Please ensure that consistent notations are used. For example, please refer to line 215 of page 10 and equation 2 of page 11. You should be able to observe inconsistent notations have been used.

6. Comparison of equations. I understand that equation 5 is augmented from equation 2. Why natural log is applied on equation 5?

7. Relevance of statistical test. Why correlation test is conducted given that your model will also capture correlation?

8. Measure for CD. I understand that Hofstede scores are not provided annually. May I know how you construct CD over the stated time period?

9. Robustness test. This section needs further elaboration. Why inclusion and exclusion of Vietnam indicate robustness test is conducted?

I hope you can address above comments carefully.

6. PLOS authors have the option to publish the peer review history of their article (what does this mean?). If published, this will include your full peer review and any attached files.

Reviewer #1: No

Reviewer #2: **Yes: **Chew Ging Lee

---

## [Author Response · Author response to Decision Letter 0]

18 Sep 2023

Dear Dr. Vincenzo Basile,

Thank you for giving us the opportunity to submit a revised draft of the manuscript “A study on the influence of perceived distance on China's inbound Tourism and the interaction of non-Economic distance: an Analysis of dynamic extended Gravity Model based on 61 countries' entry data (2004-2018)” for publication in the Journal of PLOS One. We appreciate the time and effort that you and the reviewers dedicated to providing feedback on our manuscript and are grateful for the insightful comments on and valuable improvements to our paper.

We have incorporated most of the suggestions made by the reviewers. Those changes are highlighted within the manuscript. Please see below, in blue, for a point-by-point response to the reviewers’comments and concerns. All page numbers refer to the revised manuscript file with tracked changes.

Reviewers' Comments to the Authors:

Reviewer 1

This paper focuses on understanding the impact of different types of distance perception on inbound tourism in China. The paper utilizes a dynamic expanding gravity model to analyze the effects of cultural, institutional, geographical, and economic distance on inbound tourism patterns from 2004 to 2018. The research is well-designed and well-analyzed.

Author response: Thank you very much for your recognition!

1 More control variables are needed. Relevant tourism-related variables such as population and transportation should be incorporated to account for comprehensive influencing factors.

Response: Thank you for this suggestion. It would have been interesting to explore this aspect. Because population variables were used in the calculation of economic distance variables, so population factors were not included in the selection of control variables.In addition, due to the availability of data, this paper uses the spatial distance of the two countries as a substitute variable when calculating the transportation cost. We agree that this is a potential limitation of the study. We have added this as a limitation on line 555 to 558 of page 27.

2.The endogeneity of variables needs to be considered. For instance, a decrease in economic distance could lead to an increase in tourist numbers, but an increase could also lead to a decrease in economic distance.

Response:We agree with the reviewer’s assessment. Accordingly, throughout the manuscript, we have introduced the first-order lag term of inbound tourism(LnTour L1.)as an independent variable to conduct dynamic model research by used system-GMM analysis. At the same time, the Hansen test and AR test in this paper are good, which shows that there is no endogeneity problem in the research model. It is shown on line 369 of page 18.

3.Some English writing in the article requires attention, such as adding plural to 'literature reviews'.

Response:Thank you for pointing this out. The reviewer is correct, and we have revised.

The revised text reads as follows on :

The original text:

“destinations and and origin countries.”

The revised text:

“destinations and origin countries.”

[line 42 of page 3]

The original text:

“culture and climate.”

The revised text:

“ culture, and climate.”

[line 61 of page 4]

The original text:

“ on the international tourism.”

The revised text:

“on international tourism.”

[line 84 of page 5]

The original text:

“Literature reviews”

The revised text

“Literature review”

[line 102 of page 5]

The original text:

“The factors of international tourism are still popular in previous tourism studies. The mainstream of previous studies analyzed the effects of geographical distance and economic distance on inbound tourism by Gravity Model. With the development of the international tourism, transportation and information technology, researchers found that the economic and geographical factors may only some factors on international tourism, and gave their sights on no-economic factors, such as cultural distance [8,9,12], institutional distance [10,11,13,14] and climate [16-18].”

The revised text:

“The mainstream of previous studies analyzed the effects of geographical and economic distance on inbound tourism using the Gravity Model. With the development of international tourism, transportation, and information technology, researchers found that economic and geographical factors may only be some of the factors influencing international tourism, and gave their insights on non-economic factors, such as cultural distance [8,9,12], institutional distance [10,11,13,14], and climate [16-18].”

[line 104 of page 6]

The original text:

“ tourist will choose the destination which is more stable and safer.”

The revised text:

“tourists will choose the destination which is more stable and safer.”

[line 113 of page 6]

The original text:

“the human prefer to ”

The revised text:

“the human prefers to ”

[line 123 of page 6]

The original text:

“the political stability of destination ”

The revised text:

“the political stability of destinations ”

[line 137 of page 7]

The original text:

“It is well known that”

The revised text:

“It is well-known that”

[line 149 of page 7]

The original text:

“culture is main pull factor in destination[46].”

The revised text:

“culture is the main pull factor in destination[46].”

[line 185 of page 9]

The original text:

“as the improvement of Internet and transport,”

The revised text:

“as the improvement of the Internet and transport,”

[line 197 of page 9]

The original text:

“And considers that the CD and LnTour may influence each other,”

The revised text:

“Considering that the CD and LnTour may influence each other,”

[line 335 of page 16]

4.Specific references in the article also need careful attention to format errors and omissions, requiring correction.

Response:Thank you for pointing this out. The reviewer is correct, and we have revised the format of the references according to the requirements of PLOS ONE.

The revised text reads as follows on :

“Gu X, Sheng L, Yuen C Y. Inbound Tourism, Hospitality Business, and Market Structure. Journal of Hospitality & Tourism Research. 2019; 43(8): 1326-1335.doi:10.1177/1096348019870574.”

[line 560 of page 28]

Reviewer 2

Thanks for writing this interesting paper that analyses the implications of different types of "distance" on tourism industry of China. Although this paper has strengths, it also has weaknesses.

Author response: Thank you very much for your comments on my article!

1.Sampling. Why only these 61 countries are included in this study? This study uses panel data. We should note that the choice of countries may affect the results. I hope the authors can address this comment carefully.

Response:We strongly agree with you about the importance of sample size in panel studies for the estimation of results. Through our efforts, we collected inbound tourism data for 127 countries, institutional distance data for 87 countries and cultural distance data for 73 countries. However, considering the completeness and availability of different data, this paper finally selected data from 61 countries as research samples. At the same time, considering the distribution number of countries in different regions, continents, economies and regions when selecting data, the sample countries of this paper are finally determined. On the whole, the sample countries selected in this paper have a good representation.[line 302 of page 14] And we have added this as a limitation on:line 552 of page 27.

2. Duration. Any particular reason 2019 data points are omitted since it is a year prior to Covid-19?

Response:Thank you for this suggestion. It would have been interesting to explore this aspect. However, in our study, this would not be possible. We further explain this problem in the paper.

The revised text reads as follows on :

“Affected by the COVID-2019, Chinese inbound tourism data for 2019-2022 is missing and belong to abnormal data. In addition, unwto data on inbound tourism has not been updated since 2019.

For this study, abnormal data processing has relatively little effect on the conclusions and empirical significance of this study, and will further the sample selection of this paper, resulting in a decrease in sample representation of this paper. Therefore, this paper selects variables such as Chinese inbound tourist arrivals, cultural distance and institutional distance from 2004 to 2018 for research. The specific variable definitions and data sources are shown in Table 1.”

[line 296 of page 14]

3. Typo and grammatical mistakes. I hope the authors can check this paper again to remove obvious typo and grammatical mistakes. For example, line 39 of page 3, there are duplicated "and".

Response:Thank you for pointing this out. The reviewer is correct, and we have revised.

The revised text reads as follows on :

The original text:

“destinations and and origin countries.”

The revised text:

“destinations and origin countries.”

[line 42 of page 3]

The original text:

“culture and climate.”

The revised text:

“ culture, and climate.”

[line 61 of page 4]

The original text:

“ on the international tourism.”

The revised text:

“on international tourism.”

[line 84 of page 5]

The original text:

“Literature reviews”

The revised text

“Literature review”

[line 102 of page 5]

The original text:

“The factors of international tourism are still popular in previous tourism studies. The mainstream of previous studies analyzed the effects of geographical distance and economic distance on inbound tourism by Gravity Model. With the development of the international tourism, transportation and information technology, researchers found that the economic and geographical factors may only some factors on international tourism, and gave their sights on no-economic factors, such as cultural distance [8,9,12], institutional distance [10,11,13,14] and climate [16-18].”

The revised text:

“The mainstream of previous studies analyzed the effects of geographical and economic distance on inbound tourism using the Gravity Model. With the development of international tourism, transportation, and information technology, researchers found that economic and geographical factors may only be some of the factors influencing international tourism, and gave their insights on non-economic factors, such as cultural distance [8,9,12], institutional distance [10,11,13,14], and climate [16-18].”

[line 104 of page 6]

The original text:

“ tourist will choose the destination which is more stable and safer.”

The revised text:

“tourists will choose the destination which is more stable and safer.”

[line 113 of page 6]

The original text:

“the human prefer to ”

The revised text:

“the human prefers to ”

[line 123 of page 6]

The original text:

“the political stability of destination ”

The revised text:

“the political stability of destinations ”

[line 137 of page 7]

The original text:

“It is well known that”

The revised text:

“It is well-known that”

[line 149 of page 7]

The original text:

“culture is main pull factor in destination[46].”

The revised text:

“culture is the main pull factor in destination[46].”

[line 185 of page 9]

The original text:

“as the improvement of Internet and transport,”

The revised text:

“as the improvement of the Internet and transport,”

[line 197 of page 9]

The original text:

“And considers that the CD and LnTour may influence each other,”

The revised text:

“Considering that the CD and LnTour may influence each other,”

[line 335 of page 16]

4. Interaction effect between cultural distance and institutional distance. The authors state that the mentioned interaction effect is not considered in existing studies. Why is the theoretical foundation of including this interaction effect? We should not estimate model just based on data availability.

Response:We agree with the reviewer’s assessment. Accordingly, throughout the manuscript, we have revised the rationality of the interaction effect introduction model from two aspects of theory and practice[line 204 of page 10].

5. Consistency of notations. Please ensure that consistent notations are used. For example, please refer to line 215 of page 10 and equation 2 of page 11. You should be able to observe inconsistent notations have been used.

Response:Thank you for pointing this out. The reviewer is correct, and we have revised according to the requirements of PLOS ONE.

The revised text reads as follows on :

“ (2)”

[line 242 of page 11].

6. Comparison of equations. I understand that equation 5 is augmented from equation 2. Why natural log is applied on equation 5?

Response:We have added the suggested content to the manuscript on[line 286 of page 13]. Due to the differences in the unit and value in the study, the degree of fitting and credibility of the research results will be reduced. In order to reduce the impact of the data differences on the results of the research model, this paper applies a logarithm, which can help to eliminate trends, and to avoid the difference of the unit of data, on variables Tour and GD.

7. Relevance of statistical test. Why correlation test is conducted given that your model will also capture correlation?

Response:: We agree with the reviewer’s assessment. Accordingly, throughout the manuscript, we have revised and deleted the correlation test.

8. Measure for CD. I understand that Hofstede scores are not provided annually. May I know how you construct CD over the stated time period?

Response: Thank you for pointing this out. This paper selects the newly published Hofstede culture score and uses the variance formula to determine the initial cultural distance. Then, by consulting the data, the diplomatic years between the sample country and China are determined as a data of bilateral cultural exchanges, and the inverse transformation is carried out and introduced into the model to form the cultural distance at any time.The specific calculation formula is shown on :line 248 of page 12.

9. Robustness test. This section needs further elaboration. Why inclusion and exclusion of Vietnam indicate robustness test is conducted?

Response:As suggested by the reviewer, we have revised the rationality of selecting Vietnam for robustness test from the aspects of data sources and the particularity of samples. In addition, the robustness test of this paper adopts the methods of eliminating part of samples and changing variable index, which can better test the robustness of the model. It is shown on line 422 of page 21.

Once again, we thank you for the time you put in reviewing our paper and look forward to meeting your expectations. Since your inputs have been precious, in the eventuality of a publication, we would like to acknowledge your contribution explicitly.

---

## [Decision Letter · Decision Letter 1]

5 Jan 2024

A study on the influence of perceived distance on China's inbound Tourism and the interaction of non-Economic distance: an Analysis of dynamic extended Gravity Model based on 61 countries' entry data (2004-2018)

PONE-D-23-18100R1

Dear Dr. Tang,

We’re pleased to inform you that your manuscript has been judged scientifically suitable for publication and will be formally accepted for publication once it meets all outstanding technical requirements.

Kind regards,

Wei-Ta Fang, Ph.D.

Academic Editor

PLOS ONE

Additional Editor Comments (optional):

Reviewers' comments:

Reviewer's Responses to Questions

**Comments to the Author**

1. If the authors have adequately addressed your comments raised in a previous round of review and you feel that this manuscript is now acceptable for publication, you may indicate that here to bypass the “Comments to the Author” section, enter your conflict of interest statement in the “Confidential to Editor” section, and submit your "Accept" recommendation.

Reviewer #1: All comments have been addressed

2. Is the manuscript technically sound, and do the data support the conclusions?

Reviewer #1: Yes

3. Has the statistical analysis been performed appropriately and rigorously? 

Reviewer #1: Yes

4. Have the authors made all data underlying the findings in their manuscript fully available?

Reviewer #1: Yes

5. Is the manuscript presented in an intelligible fashion and written in standard English?

Reviewer #1: Yes

6. Review Comments to the Author

Reviewer #1: The author(s) did a comprehensive and detailed job and addressed all the concerns I had regarding the manuscript.  In its current form the paper can be considered for publication but I would suggest that the author(s) will run the manuscript through technical editing, especially the format of reference and figure, to ensure that it is in good publication format.

7. PLOS authors have the option to publish the peer review history of their article (what does this mean?). If published, this will include your full peer review and any attached files.

Reviewer #1: No
